# Development of Ratiometric Fluorescence Sensors Based on CdSe/ZnS Quantum Dots for the Detection of Hydrogen Peroxide

**DOI:** 10.3390/s19224977

**Published:** 2019-11-15

**Authors:** Hong Dinh Duong, Jong Il Rhee

**Affiliations:** School of Chemical Engineering and Research Center for Biophotonics, Chonnam National University, Yong-Bong Ro 77, Gwangju 61186, Korea; zink1735@gmail.com

**Keywords:** ratiometric fluorescence QD membrane, CdSe/ZnS QDs, redox reaction, hydrogen peroxide, α-ketobutyrate

## Abstract

In this study, carboxyl group functionalized-CdSe/ZnS quantum dots (QDs) and aminofluorescein (AF)-encapsulated polymer particles were synthesized and immobilized to a sol–gel mixture of glycidoxypropyl trimethoxysilane (GPTMS) and aminopropyl trimethoxysilane (APTMS) for the fabrication of a hydrogen peroxide-sensing membrane. CdSe/ZnS QDs were used for the redox reaction of hydrogen peroxide (H_2_O_2_) via a reductive pathway by transferring electrons to the acceptor that led to fluorescence quenching of QDs, while AF was used as a reference dye. Herein, the ratiometric fluorescence intensity of CdSe/ZnS QDs and AF was proportional to the concentration of hydrogen peroxide. The fluorescence membrane (i.e., QD–AF membrane) could detect hydrogen peroxide in linear detection ranges from 0.1 to 1.0 mM with a detection limit (LOD) of 0.016 mM and from 1.0 to 10 mM with an LOD of 0.058 mM. The sensitivity of the QD–AF membrane was increased by immobilizing horseradish peroxidase (HRP) over the surface of the QD–AF membrane (i.e., HRP–QD–AF membrane). The HRP–QD–AF membrane had an LOD of 0.011 mM for 0.1–1 mM H_2_O_2_ and an LOD of 0.068 mM for 1–10 mM H_2_O_2_. It showed higher sensitivity than the QD–AF membrane only, although both membranes had good selectivity. The HRP–QD–AF membrane could be applied to determine the concentration of hydrogen peroxide in wastewater, while the QD–AF membrane could be employed for the detection of α-ketobutyrate.

## 1. Introduction

Hydrogen peroxide (H_2_O_2_) is usually one of the final products in many biological processes after biological substrates react with oxygen through the catalysis of some oxidases [1]. The amount of substrate used can be measured indirectly via the amount of H_2_O_2_ produced [2,3,4]. H_2_O_2_ also acts as a reactive oxygen species when it is excessively present above the tolerance threshold of biological organisms. Excessive amounts of H_2_O_2_ are associated with cell damage and other disorders [5]. H_2_O_2_ is a chemical compound that is widely used in the bleaching industry [6], in wastewater treatment [7], in medical disinfectants, and other applications in human life [8,9]. Therefore, the detection of H_2_O_2_ and its control in each field are needed for safe human life. Among numerous analytical methods published for the detection of H_2_O_2_, fluorescence methods are the most preferred ones due to their high sensitivity and selectivity [1,10]. Many fluorescence probes showed strong affinity to react with H_2_O_2_ and they were applied for H_2_O_2_ detection in living cells [11,12,13,14,15]. However, these probes are commonly used by adding them to aqueous samples, and they cannot be reused [16,17,18,19,20]. Scientists are looking for new ways to convert fluorescence probes into optical sensors for reuse. Among fluorescence probes, quantum dots (QDs) are theoretically excellent probes for detecting H_2_O_2_ due to the possible photo-induced electron transfer mechanism (PET) [21] or fluorescence resonance energy transfer (FRET) process [22]. However, optical H_2_O_2_ sensors using QD-based sensing membranes are yet to be reported. H_2_O_2_-sensing membranes prepared with other common probes are also rarely published [1]. Recently, optical H_2_O_2_ sensors were presented as sensing membranes containing fluorescence dyes such as ruthenium complex [23,24,25], HP green fluorescence probe [26], and europium tetracycline complex [27] entrapped in some types of polymers. However, the single fluorescence intensity-based sensing technique is influenced by the local distribution of dye and drifts of light source and detectors. Alternatively, ratiometric fluorescence detection with two signals offers built-in corrections to minimize these factors and enable accurate readouts.

Therefore, in this study, ratiometric fluorescence H_2_O_2_ sensors were developed using CdSe/ZnS QDs as a detection dye and aminofluorescein (AF) as a reference dye (Scheme 1). For the development of some H_2_O_2_-sensing membranes, CdSe/ZnS QDs were synthesized and AF was captured in melamine-formaldehyde (MF) particles (AF@MF particles). CdSe/ZnS QDs and AF@MF particles were immobilized into a sol–gel matrix of glycidoxypropyl trimethoxysilane (GPTMS) and aminopropyl trimethoxysilane (APTMS) as a supporting material. This membrane immobilized with CdSe/ZnS QDs and AF@MF particles (i.e., QD–AF membrane) showed high sensitivity in detecting different H_2_O_2_ concentrations through the fluorescence-quenching efficiency of CdSe/ZnS QDs. The sensitivity of the QD–AF membrane was improved after immobilizing horseradish peroxidase (HRP) on the QD–AF membrane (i.e., HRP–QD–AF membrane). The operating mechanism, advantages, and limitations of these H_2_O_2_-sensing membranes are described in detail in this study.

## 2. Materials and Methods

### 2.1. Materials

Cadmium acetate, selenium, zinc acetate, hexamethyldisilathiane ((TMS)_2_S), trioctylphosphin oxide (TOPO), trioctylphosphine (TOP), ethyl cellulose (EC), mercaptopropionic acid (MPA), 4-dimethylamino pyridine (DMAP), 1-hexadecylamine, stearic acid, 3-glycidoxypropyl trimethoxysilane (GPTMS), 3-aminopropyltrimethoxysilane (APTMS), aminofluorescein (AF), melamine, paraformaldehyde, horseradish peroxidase (HRP), hydrogen peroxide, and sodium α-ketobutyrate were purchased from Sigma-Aldrich Chemical Co. (Seoul, Korea). *N*,*N*-Dimethylformamide (DMF), dimethylsulfoxide (DMSO), methanol, ethanol, and chloroform were obtained from Honeywell International Inc. Burdick & Jackson (Ulsan, Korea). Other chemicals such as hydrochloric acid, sodium hydroxide, sodium phosphate (mono- and dibasic), sodium chloride, and phosphate buffer were of analytical grade and used without further purification.

### 2.2. Synthesis of CdSe/ZnS QDs

The synthesis of CdSe/ZnS core/shell QDs was based on methods of Qu and Peng [28] and Gaunt et al. [29] with slight modifications. Cadmium acetate dehydrate (0.6 mM, 147 mg) and stearic acid (2.13 mM, 607 mg) were loaded into a 50-mL three-neck flask and heated to 150 °C under vacuum conditions until a colorless liquid was obtained. Hexadecylamine (1.94 g) and TOPO (2.2 g) were added to the flask after cooling to room temperature. The mixture was then degassed with a pump and heated to 120–150 °C under vacuum. The reaction vessel was then filled with nitrogen gas and heated to 310–320 °C; at this point, a solution of selenium (211 mg) in TOP (2.5 mL) was rapidly injected into the vigorously stirred reaction mixture. The solution was heated for 25 s before removing the flask from the heating mantle and then allowing it to cool to room temperature. The resulting CdSe nanoparticles were purified by dissolving the reaction mixture in chloroform, followed by precipitation with an equal volume of methanol. In the next step, these purified CdSe particles were used to synthesize the CdSe/ZnS core/shell QDs. A mixture of hexadecylamine (2 g) and TOPO (2.5 g) was loaded into a 50-mL three-neck flask and then degassed and heated to 180 °C. At 180 °C, the purified CdSe particles dispersed, and 2.0 mL of chloroform was added to this solution. After the chloroform was completely pumped out, the flask was filled with nitrogen gas. The temperature of the reaction was then increased to 180–185 °C. A mixture of zinc acetate (54 mg) and (TMS)_2_S (0.05 mL) dissolved in 1.0 mL of TOP was then injected dropwise for 5–10 min. After the injection, the mixture was stirred for 1 h at 180–185 °C. Fluorescence quantum yield of the synthesized CdSe/ZnS QDs was calculated using the following equation [30], where rhodamine 6G was used as standard reference:(1)$$\phi_{x}=\phi_{\mathrm{std}}\times \frac{F_{x}}{F_{\mathrm{std}}}\times \frac{f_{\mathrm{std}}\left(\lambda_{\mathrm{ex}}\right)}{f_{x}\left(\lambda_{\mathrm{ex}}\right)}\times \frac{n_{x}^{2}}{n_{\mathrm{std}}^{2}},$$
where F is the area under the emission spectrum, f is absorption factor at a given excitation wavelength of sample or standard, n is the refractive index of solvent, and Φ is fluorescence quantum yield. The subscript “x” and “std” represent sample and standard.

For hydrophilic surfactant-capped CdSe/ZnS QDs, MPA-coated QDs were prepared following published protocols [31,32,33] with slight modifications. Fluorescence emission spectra of CdSe/ZnS QDs were determined with a fluorescence spectrophotometer (Model: F-4500, Hitachi Co., Tokyo, Japan). TEM images were acquired with an FEI Tecnai G^2^ twin system (FEI Co., Hillsboro, OR, USA).

### 2.3. Synthesis of Aminofluorescein (AF)-Encapsulated Melamine-formaldehyde (MF) Particles

Melamine-formaldehyde (MF) solution was prepared according to the protocol of Wu et al. [34]. Briefly, 2.6 g of melamine was firstly mixed with 3.7 g of paraformaldehyde and 50 mL of water. The mixture was heated at 50 °C for 40 min with magnetic stirring. The transparent MF solution was subsequently filtered with two layers of filter paper and stored at 4 °C until further use. In the next step of the synthesis of AF-encapsulated MF spherical particles (AF@MF particles), 1 mL of 10 mM AF in DMSO was mixed with 10 mL of the prepared transparent MF solution and then mixed with 22 mL of diluted hydrochloric acid (0.4 mM). Subsequently, the mixture was heated at 100 °C in a boiling water bath for 30 min to obtain a suspension of AF@MF particles. Products were separated and washed several times with distilled water by centrifugation and redispersion cycles. They were then dried at 60 °C for 12 h in an oven. Fluorescence spectra of AF@MF particles were measured with a fluorescence spectrometer (Model: F-4500, Hitachi Co., Tokyo, Japan).

### 2.4. Preparation of H_2_O_2_-sensing Membranes

#### 2.4.1. QD–AF Membrane

Firstly, a sol–gel solution was prepared according to the procedure described in our previous report [35]. Sol–gel GPTMS/APTMS (GA) was prepared by hydrolyzation and polymerization of the mixtures of GPTMS and APTMS in ethanol solvent with volumetric ratio of 25/6.5/68.5 in the presence of 40 µL of hydrochloric acid for at least 2 h at room temperature. The sol–gel GA membrane containing hydrophilic MPA-coated CdSe/ZnS and AF@MF particles was prepared by dissolving the precipitated hydrophilic CdSe/ZnS QDs (from 1.8 mL of MPA-QDs in DMF) with 300 µL of the sol–gel GA matrix and then mixing with 3 mg of AF@MF particles in 100 µL of the sol–gel GA matrix. The mixture was aged at room temperature for 3 h. Then, 10 µL of the mixture was spread on the bottom of one well of a 96-well microtiter plate (NUNC Co., Copenhagen, Denmark) and dried at 60 °C for 24 h.

#### 2.4.2. HRP–QD–AF Membrane

After 20 µL of 10 wt.% EC in ethanol and water (9:1 *v*/*v*) was mixed with 10 units (U) of HRP dissolved in 0.1 M phosphate buffer (pH 7.0), the mixture was used to coat the QD–AF membrane in a 96-well microtiter plate. The QD–AF membrane immobilized with HRP (i.e., HRP–QD–AF membrane) was incubated at 4 °C for 24 h. The presence of a small amount of aqueous HRP solution led to the formation of the EC membrane involving enzyme entrapment. Surface morphologies of QD–AF and HRP–QD–AF membranes were examined by atomic force microscopy (AFM).

### 2.5. Fluorescence Measurements

Concentrations of H_2_O_2_ to be measured were in the range of 0.1–10 mM. Data were collected from the fluorescence intensity of the H_2_O_2_-sensing membrane at two emission wavelengths (*λ_em_* = 500 nm and *λ_em_* = 560 nm) with an excitation wavelength of 400 nm (*λ_ex_* = 400 nm). Fluorescence spectra for the detection of H_2_O_2_ were measured with a multifunctional fluorescence microplate reader (Safire^2^, Tecan Austria GmbH, Wien, Austria). The immobilization efficiency of HRP in the HRP–QD–AF membrane was calculated by dividing the amount of immobilized HRP by the total amount of HRP used for immobilization. The amount of the immobilized HRP was determined by subtracting the amount of unimmobilized HRP from the total amount of HRP used. Unimmobilized HRP was separated from immobilized HRP in one well by washing several times with 1.0 mL of 10 mM phosphate buffer (pH 7.4). Protein concentrations of the washed, unimmobilized HRP were determined with the Bradford method. Optimization of HRP amount for immobilization was performed with 1, 5, 10, 15, and 20 units (U). Sensitivities of the HRP–QD–AF membranes immobilized with different amounts of HRP were evaluated based on the slope value (sensitivity index (SI)), i.e., the ratio of fluorescence intensities at two emission wavelengths (λ_em_ = 500 nm and λ_em_ = 560 nm) considering H_2_O_2_ concentration. Kinetic parameters (K_m_ and V_max_) of the immobilized HRP were determined from the Lineweaver–Burk plot based on the ratio of fluorescence intensities at λ_em_ = 500 and λ_em_ = 560 nm.

Reversibility of the H_2_O_2_-sensing membrane was performed at 0.1, 1.0, and 10 mM H_2_O_2_. The H_2_O_2_-sensing membrane was measured in a sequence of H_2_O_2_ concentrations (from low to high concentrations), and the measurement cycle was repeated. A multifunctional fluorescence microplate reader was set for fluorescence measurements against time with an interval of 30 s during 10 min.

Effects of pH and temperature on H_2_O_2_ measurements were investigated. Solutions of 1 mM H_2_O_2_ with pH in the range of pH 5.0 to pH 9.0 were exposed to the H_2_O_2_-sensing membrane. The H_2_O_2_-sensing membrane was also tested with different temperatures (25, 30, 33, 35, 37, and 40 °C) for H_2_O_2_ concentrations ranging from 0.1 to 10 mM. The long-term stability of the H_2_O_2_-sensing membrane was evaluated by determining its repeatability by measuring the fluorescence intensity at various H_2_O_2_ concentrations after a number of measurements.

Response of the QD–AF membrane to H_2_O_2_ was used to detect α-ketobutyrate according to the following reaction:α-ketobutyrate + H_2_O_2_ → propionate + H_2_O.

Residual amounts of H_2_O_2_ were expressed via fluorescence quenching of the QD–AF membrane after a fixed amount of H_2_O_2_ (10 mM) reacted with 1 mL of different concentrations of α-ketobutyrate. These quenched fluorescence intensities corresponded to concentrations of α-ketobutyrate. Concentrations of α-ketobutyrate were also colorimetrically determined by fluorescence quenching of the QD–AF membrane. The QD–AF membrane was exposed to solutions resulting from the reaction of H_2_O_2_ with different concentrations of α-ketobutyrate. Changes in the color of the QD–AF membrane were obtained using a microscopic fluorescence camera (AM4115T-GRFBY Dino-Lite Edge: *λ_ex_* = 470 nm and *λ_em_* = 510 nm, AnMo Electronics Co., Taipei, Taiwan) placed in a homemade black chamber.

Concentrations of H_2_O_2_ in artificial wastewater were also determined using the HRP–QD–AF membrane. The artificial wastewater solution containing 2.5 mM CaCl_2_, 45 mM NaCl, 3.5 mM KH_2_PO_4_, 3.5 mM K_2_HPO_4_, 2.5 mM NaHCO_3_, 1 mM MgSO_4_, 2.5 mM Na_2_SO_4_, and H_2_O_2_ at different concentrations (0.1–10 mM) was prepared.

### 2.6. Ratiometric Method

The ratiometric calculation for the H_2_O_2_-sensing membrane was based on the ratio (R) of fluorescence intensities at two emission wavelengths (*λ_em_* = 500 nm (FI*_500_*) and *λ_em_* = 560 nm (FI*_560_*)) as follows:R = FI_560_/FI_500_.

Normalized ratiometric fluorescence intensity was calculated by dividing the R by the R_max_, i.e., the maximum value of R.

### 2.7. Data Analysis

Differences in the ratiometric fluorescence intensity of H_2_O_2_-sensing membranes at different levels of oxidant and temperatures were assessed by one-way analysis of variance (ANOVA). A significant difference between samples was accepted at *p* < 0.05. All statistical tests were performed using InStat software vers.3.01 (GraphPad Software Inc., San Diego, CA, USA).

## 3. Results and Discussion

### 3.1. Properties of CdSe/ZnS QDs and AF@MF Particles

CdSe/ZnS QDs with high quantum yield (QY of 60%) were synthesized with an emission wavelength of 590 nm, a full width at half maximum of the emission spectrum (FWHM) of 30 nm, and a size of around 7 nm (Figure 1a). After ligand exchange with carboxyl groups (MPA) on the surface of the synthesized CdSe/ZnS QDs, their fluorescence emission wavelength did not change significantly. However, the QY of MPA-coated CdSe/ZnS QDs was decreased to about 50% of its initial value.

The AF@MF particles had a size of about 1 µm (Figure 1a). The band edge of the emission wavelength of the AF@MF particles was 530 nm (Figure 1b). As shown in our previous studies [35,36], sol–gel GA is a good supporting material for tightly capturing the hydrophilic CdSe/ZnS QDs and AF into its matrix via static interaction or covalent binding of amine groups of the sol–gel GA to carboxyl groups of the hydrophilic CdSe/ZnS QDs or epoxy groups of the sol–gel GA to amine groups of AF, respectively, without influencing optical properties of CdSe/ZnS QDs or AF.

### 3.2. Characterization of the QD–AF Membrane

Hydrophilic CdSe/ZnS QDs and AF@MF particles were combined with the sol–gel GA matrix to fabricate QD–AF membranes on the bottom of wells of a 96-well microtiter plate. Outcome signals from this QD–AF membrane indicated that emission wavelengths of CdSe/ZnS QDs and AF@MF particles were shifted to shorter wavelengths (i.e., 560 nm for CdSe/ZnS QDs and 500 nm for AF@MF particles) with an excitation wavelength of 400 nm (λ_ex_ = 400 nm). As shown in Figure 2a, the fluorescence intensity of the QD–AF membrane at λ_em_ = 560 nm decreased significantly with increasing H_2_O_2_ concentrations, while the fluorescence intensity of the membrane at λ_em_ = 500 nm was stable during measurements. Based on ratiometric fluorescence intensities at λ_em_ = 500 and 560 nm, the linear detection range of the QD–AF membrane included a range of 0.1–1.0 mM H_2_O_2_ with a regression coefficient value of *r^2^*_0.1–1_ = 0.978. It also included a range of 1.0–10 mM H_2_O_2_ with a regression coefficient value of *r^2^*_1–10_ = 0.972. The detection limit (LOD) was 0.016 mM for the range of 0.1–1 mM H_2_O_2_ and 0.058 mM for the range of 1–10 mM H_2_O_2_.

Fluorescence quenching of CdSe/ZnS QDs by H_2_O_2_ could be seen by naked eyes through color changes of the QD–AF membrane exposed to different H_2_O_2_ concentrations and under LED470 irradiation (Figure 2b). As shown in Figure 2c, the fluorescence intensity (FI) of the reference dye such as AF was remarkably lower than the FI of CdSe/ZnS QDs. This is an advantage for ratiometric calculation between CdSe/ZnS QDs (FI_560_) and AF@MF particles (FI_500_) to get a high slope value from the linear equation. Moreover, adding a certain amount of a reference dye could affect activities of indicators in some cases. Therefore, the use of a small amount of AF was preferred in this work. Advantage of using low (3 mg of AF@MF particles) and high (6 mg of AF@MF particles) amounts of the reference dye (AF) could be validated through data shown in Figure 2c. As shown in Figure 2c, the slope value using a small amount of AF@MF particles was higher than the slope value using a large amount of AF@MF particles in concentration ranges of 0.1–1 mM and 1–10 mM for H_2_O_2_. The sensitivity (SI = 0.209 and 0.056) of the QD–AF membrane with a small amount of AF@MF particles was higher than the sensitivity (SI = 0.0116 and 0.046) of the QD–AF membrane with a large amount of AF@MF particles.

The fluorescence quenching of CdSe/ZnS QDs could result from several reactions that simultaneously or serially occur in the QD–AF membrane depending on environmental conditions. H_2_O_2_ is either an oxidizing or reducing agent depending on the pH of the environment. In our pre-test, solutions of H_2_O_2_ in the concentration range of 0.1–10 mM were weak acidic solutions with pH values varying from pH 5.12 to 5.2. Under this condition, H_2_O_2_ exhibited as an oxidizer. It means that H_2_O_2_ could receive electrons from an electron donor.

In this case, CdSe/ZnS QDs exhibited as excellent electron donors upon photo-excitation of CdSe/ZnS QDs (see Equation (4) = Equation (2) + Equation (3))
QDs – 2e^−^ + *hv* → QDs^*^.(2)
H_2_O_2_ + 2e^−^ + 2H^+^ → 2H_2_O.(3)
QDs + H_2_O_2_ + 2H^+^ + *hv* → QDs^*^ + 2H_2_O.(4)

The photo-induced electron transfer between electron donation of CdSe/ZnS QDs and electron acceptance of H_2_O_2_ led to a decrease in the fluorescence intensity of QDs. The fluorescence quenching of QDs in the presence of H_2_O_2_ can also be identified by other methods. This is a new cooperative transition provided by dimer formation between species in the excited electron and another species in the ground state, leading to a movement in optical emission toward lower energy [37]. Bond formation at the surface (change in local charge densities or formation of surface states) can affect the quantum efficiency of QDs [38].

According to studies of Brus et al. [39], QDs can highly recover fluorescence photophysics, in which the photo-generated electron–hole pair recombines by transferring its energy to the strongly coupled, resident third carrier, leading to non-emission of an ionized nanocrystal. When the nanocrystal is neutralized via a second photo-ionization event, the emission is restored. In this study, the QD–AF membrane showed high reproducibility (Figure 3a) with small values of relative standard deviation (RSD): 0.3% at 0.1 mM, 2.6% at 1 mM, and 0.1% at 10 mM H_2_O_2_.

One of the most important parameters that affects the accuracy of a sensor is a specific reaction for selective detection of analyte. In this study, the QD–AF membrane was exposed to various oxidizing agents such as sodium hypochlorite (NaClO), chloramine T (ChT), and potassium permanganate (KMnO_4_) at different concentrations to evaluate the specificity of this QD–AF membrane. As shown in Figure 3b, the response of the QD–AF membrane to different H_2_O_2_ concentrations was absolutely contrastable with that of the membrane when it was exposed to other oxidizing agents. In the case of H_2_O_2_, the normalized ratiometric fluorescence intensity decreased with increasing concentrations of H_2_O_2_ because of the quenching effect of CdSe/ZnS QDs and the stability of the reference dye (AF). The response of the QD–AF membrane showed an upward tendency with increasing concentrations of oxidizing agents since the quenching effect of CdSe/ZnS QDs did not occur, while the fluorescence intensity of CdSe/ZnS QD increased with increasing concentrations of oxidizing agents. These results indicate that the QD–AF membrane is a sensitive and selective membrane to H_2_O_2_ based on the fluorescence quenching effect of CdSe/ZnS QDs. The interference of these oxidizing agents during measurement of H_2_O_2_ was also investigated. The QD–AF membrane was exposed to different concentrations of H_2_O_2_ and a fixed amount of each oxidizing agent. Its response is shown in Figure 3c. There was no effect of 1 mM ChT on the measurement of H_2_O_2_ (*p* = 0.2087). However, the addition of 1 mM NaClO to different concentrations of H_2_O_2_ affected the sensitivity of the QD–AF membrane according to data collected from statistical analysis (*p* < 0.001). In theory, H_2_O_2_ and NaClO can react with each other when H_2_O_2_ becomes a reducing agent in the alkaline solution. However, acidic solutions of H_2_O_2_ do not create favorable conditions for the occurrence of this reaction. Therefore, the reason for the effect of NaClO on the reaction for H_2_O_2_ measurement is still unclear. When 1 mM KMnO_4_ was added, the fluorescence quenching of CdSe/ZnS QDs was not identified. KMnO_4_ had no reaction with H_2_O_2_ under acidic conditions. The dark-brown color of the KMnO_4_ solution can either absorb energy from the illumination of the light source or prevent the photo-excited state of CdSe/ZnS QDs from reacting with H_2_O_2_.

As mentioned above, the QD–AF membrane showed good properties for H_2_O_2_ detection. However, the membrane also showed its limitation because its lifetime was not very long in terms of sensitivity at high concentrations of H_2_O_2_ (Figure 4). According to the theory of Wosnick [40], the sol–gel GA matrix is a fluorescent semiconductor polymer that can absorb a photon to generate an exciton. The migration of the exciton along the polymer backbone plays the role of charge carrier in an electrically conducting polymer. Moreover, the exciton tends to migrate toward low-energy trapping sites. These trapping sites can serve as energy sinks in the polymer. CdSe/ZnS QDs conjugated with polymer are traps or they generate trapping sites. These sites are used for trapping electron energy from the outside of the polymer. Due to photon absorption of the sol–gel GA membrane during the measurement, an electron increase in the p- and n-type regions was produced in the polymer membrane after many uses. At that time, H_2_O_2_ not only received electrons upon photo-excitation of CdSe/ZnS QDs, but also somewhat interacted with free electrons of the n-type region of the polymer membrane, which degraded the quenching efficiency of CdSe/ZnS QDs at high concentrations of H_2_O_2_. Fortunately, this situation was not observed in low concentrations of H_2_O_2_ since the sensitivity of the QD–AF membrane was stable after a long time of use. Compared with that of the sol–gel GA membrane, the rapid response of the excited CdSe/ZnS QDs to H_2_O_2_ in the redox reactions is possible.

### 3.3. Characterization of the HRP–QD–AF Membrane

The sensitivity of the QD–AF membrane could be improved at low concentrations of H_2_O_2_ by immobilizing HRP onto the membrane to catalyze the redox reaction of H_2_O_2_. The HRP–QD–AF membrane included two layers. The lower layer was the QD–AF membrane and the upper layer was a layer of HRP immobilized on the EC matrix. EC was used as a supporting material for enzyme immobilization, the capability of which was confirmed in our previous study [41]. Morphologies of the first layer and the second layer are presented in Figure 5. In comparison with our previous study [35], the surface of the QD–AF membrane was rougher than that of the QD membrane only. This was due to the presence of AF@MF particles (Figure 5a). Herein, the surface mean roughness (Ra) of 1.53 nm and the root-mean-square roughness (Rq) of 2 nm were obtained, while the surface of the second layer immobilized with HRP had larger roughness, i.e., Ra of 11.99 nm and Rq of 14.24 nm. The HRP–QD–AF membrane seemed to swell after enzyme immobilization (Figure 5b), but it was quite stable during measurements.

The operating mechanism of the HRP–QD–AF membrane was based on the fluorescence quenching effect of the QD–AF membrane in the presence of H_2_O_2_ as mentioned in Section 3.2. However, in this case, HRP contributed to the fluorescence quenching of CdSe/ZnS QDs. HRP is oxidized by using H_2_O_2_ as an oxidizing agent (Equation (5)). After it was oxidized, the HRP_(ox)_ continued the turnover of charge transfer among the involved moieties upon photo-excitation of QDs (Equation (6)), leading to fluorescence quenching of QDs.
HRP_(red)_ + H_2_O_2(ox)_ + 2H^+^ → HRP_(ox)_ + 2H_2_O.(5)
HRP_(ox)_ + QDs + *hv* → HRP_(red)_ + QDs^*^.(6)

According to data collected from the Bradford protein assay, the immobilization efficiency of HRP on the EC matrix above the QD–AF membrane was quite similar when initial amounts of HRP were 5, 10, and 15 U. The use of smaller or larger amounts of HRP did not give better results. Approximately 22.6%, 45.6%, 47.9%, 49.3%, and 20.1% of the used HRP was immobilized with initial amounts of 1 U, 5 U, 10 U, 15 U, and 20 U, respectively. That is, amounts of the immobilized HRP corresponded to 0.22 U, 2.28 U, 4.79 U, 7.39 U, and 4.03 U, respectively (Figure 6a).

The response of the HRP–QD–AF membranes immobilized with different amounts of HRP indicated that the membrane immobilized with 10 U of HRP showed the highest sensitivity among these membranes immobilized with various amounts of HRP (slope value, SI_10U_ = 4.98) (Figure 6b). As per the theory, a large amount of immobilized enzyme may lead to a narrow detection range or cause a transfer obstruction of analyte to contact with the QD–AF membrane, while a small amount of immobilized enzyme can result in a slow reaction, a slow response, and low sensitivity of the HRP–QD–AF membrane. Therefore, we preferred to use 10 U for the fabrication of the HRP–QD–AF membrane.

Optical properties of the HRP–QD–AF membrane did not change after enzyme immobilization (Figure 7). Fluorescence quenching of the HRP–QD–AF membrane with H_2_O_2_ concentrations was found at an emission wavelength (λ_em_) of 560 nm. Moreover, the sensitivity of the HRP–QD–AF membrane increased considerably at low concentrations of H_2_O_2_ since the slope value (SI = 0.365) in the linear detection range of H_2_O_2_ from 0.1 to 1.0 mM was higher than that of the QD–AF membrane (SI = 0.302 in Figure 2a). The limit of detection (LOD) of the HRP–QD–AF membrane was 0.011 mM for H_2_O_2_ in the range of 0.1 to 1.0 mM. It was 0.068 mM for H_2_O_2_ in the range of 1 to 10 mM. In addition, the activity of HRP immobilized on the EC matrix was evaluated via Michaelis–Menten kinetics. Kinetic parameters were calculated from the ratio of two emission fluorescence intensities at λ*_em_* = 500 and 560 nm. The maximal reaction rate (V_max_) of 0.012 mM/min and Michaelis–Menten constant (K_m_) of 0.206 mM were obtained from the Lineweaver–Burk plot. Immobilization of HRP improved the fluorescence quenching of the QD–AF membrane. The response time of the HRP–QD–AF membrane was approximately t_95_ = 5.5–8.5 min.

After enzyme immobilization, the reproducibility of the HRP–QD–AF membrane was well maintained when the sensing membrane was exposed to different H_2_O_2_ concentrations and the measurement cycle was repeated (Figure 8a). The RSD value was 0.77% for 0.1 mM H_2_O_2_, 1.74% for 1 mM H_2_O_2_, and 0.6% for 10 mM H_2_O_2_. The upper layer of the HRP–QD–AF membrane containing polymer EC and enzyme HRP was not a barrier to the interaction between H_2_O_2_ and CdSe/ZnS QDs in the sensing membrane. In Figure 8b, the HRP–QD–AF membrane showed the same tendency for H_2_O_2_ and other oxidizing agents as compared to the QD–AF membrane in Section 3.2. The fluorescence quenching effect of CdSe/ZnS QDs was not observed in the presence of different concentrations of oxidizing agents such as sodium hypochlorite (NaClO), chloramine T (ChT), or potassium permanganate (KMnO_4_). When the HRP–QD–AF membrane was exposed to different concentrations of H_2_O_2_ in the presence of 1 mM of each oxidizing agent (Figure 8c), 1 mM ChT did not affect the measurement of H_2_O_2_ (*p* = 0.2925) whereas 1 mM NaClO or 1 mM KMnO_4_ showed the same situation as mentioned in Section 3.2. NaClO affected the sensitivity of the HRP–QD–AF membrane (*p* = 0.0274) to H_2_O_2_. KMnO_4_ either absorbed energy from the light illumination or prevented the excitation state of CdSe/ZnS QDs from getting reactions with H_2_O_2_.

Effects of temperature on the HRP–QD–AF membrane are shown in Figure 9. The sensitivity of the HRP–QD–AF membrane to H_2_O_2_ in the range of 0.1–1.0 mM did not change significantly (*p* > 0.05) when it was exposed in the temperature range of 25–40 °C. However, for H_2_O_2_ in the high concentration range (1–10 mM), the sensitivity of the HRP–QD–AF membrane changed significantly (*p* < 0.001) when the temperature was over 33 °C (Figure 9a). In fact, CdSe/ZnS QDs are very sensitive to temperature. Their electron energy decreases with increasing temperature [42]. Herein, the temperature varied in a narrow range. Thus, effects of temperature on fluorescence quenching were not clear. The activity of HRP was increased with increasing temperature. Thus, the high reaction rate of HRP at high temperature and high concentration of H_2_O_2_ could be improved, and the sensitivity of the HRP–QD–AF membrane was increased.

When the HRP–QD–AF membrane was exposed to different pHs, the ratiometric fluorescence intensity (FI_560_/FI_500_) was strongly decreased at low pHs. It was increased with increasing pH to alkaline medium as shown in Figure 9b. H_2_O_2_ acts as an oxidizing agent in acidic solutions. However, it acts as a reducing agent in alkaline solutions. The operating mechanism of the HRP–QD–AF membrane in acidic solutions was mentioned above. However, in alkaline solutions, H_2_O_2_ acted as a reducing agent (Equation (7)). The electrons from Equation (7) can spin-pair with electrons of CdSe/ZnS QDs upon QD photo-excitation to form a covalent bond. Such a covalent-bond interaction can lower the energy of the charge transfer state and cause a lower fluorescence intensity of CdSe/ZnS QDs [37]. This mechanism might not be strongly operated by the QD membrane, leading to low quenching efficiency of CdSe/ZnS QDs.
H_2_O_2_ – 2e^−^ + 2OH^−^ → 2H_2_O + O_2_.(7)

The lifetime of the HRP–QD–AF membrane was determined by several factors mentioned in Section 3.2 (Figure 10). Due to energy accumulation and formation of the sol–gel GA layer after long-term photon absorption, the interaction of CdSe/ZnS QDs with H_2_O_2_ was influenced by the interaction of the sol–gel GA membrane with H_2_O_2_ at high concentrations of H_2_O_2_, leading to a depression of the fluorescence quenching efficiency of CdSe/ZnS QDs and the sensitivity of the HRP–QD–AF membrane. In addition, functional groups on the surface of CdSe/ZnS QDs and components of QDs could be oxidized by H_2_O_2_, thus affecting on fluorescence quenching of QDs. However, the sensitivity of the HRP–QD–AF membrane was stable at the low concentration range of H_2_O_2_ for a long time.

### 3.4. Applications of QD–AF and HRP–QD–AF Membranes

#### 3.4.1. Detection of α-ketobutyrate Using the QD–AF Membrane

α-Ketobutyrate is an active intermediate in organic synthesis and biological metabolism. It iswidely used in biology [43], medicine [44], and food additive productions [45,46,47]. Generally, α-ketobutyrate is determined by high-pressure liquid chromatography [45,48,49]. No new optical method for determination of α-ketobutyrate was published until now. Based on the reaction of α-ketobutyrate with H_2_O_2_ and the fluorescence quenching of the CdSe/ZnS QDs in the presence of H_2_O_2_, α-ketobutyrate was indirectly determined through the response of the QD–AF membrane in this study. Figure 11a shows fluorescence quenching results of the QD–AF membrane after exposure to solutions having different concentrations of α-ketobutyrate and a fixed concentration of H_2_O_2_ (10 mM). The fluorescence quenching increased with decreasing concentrations of α-ketobutyrate. The calibration curve shown in Figure 11a revealed that the linear detection range of α-ketobutyrate was from 0.2 to 2 mM with an LOD of 0.035 mM, and from 2 to 20 mM with an LOD of 1.5 mM. In addition, the concentration of α-ketobutyrate could be determined by colorimetric measurements as shown in Figure 11b. Color changed from red to dark brown when the concentration of α-ketobutyrate was decreased from 100 mM to 0.2 mM.

#### 3.4.2. Determination of H_2_O_2_ in Artificial Wastewater Using the HRP–QD–AF Membrane

The recovery capability of the HRP–QD–AF membrane was quite good for determining H_2_O_2_ in wastewater. Due to the operating mechanism, the HRP–QD–AF membrane in the presence of H_2_O_2_ was quite typical. This membrane was not much influenced by the presence of some components in artificial wastewater. Recovery percentage of the HRP–QD–AF membrane was very high (Figure 12), ranging from 99.8% to 107% at low concentrations (0.1–1.0 mM) of H_2_O_2_ and from 107% to 111.5% at high concentrations (1–10 mM) of H_2_O_2_.

For the detection of hydrogen peroxide, some techniques using quantum dots (QDs) were reviewed as shown in Table 1. Among these QD-based H_2_O_2_ sensors, the ratiometric fluorescence H_2_O_2_ sensor in this work exhibited good analytical performance in terms of sensitivity and stability. In particular, many samples can also be measured simultaneously by using a 96-well microtiter plate with immobilized H_2_O_2_-sensing membranes.

## 4. Conclusions

CdSe/ZnS QDs and aminofluorescein (AF)-encapsulated melamine-formaldehyde particles were successfully employed to fabricate a ratiometric fluorescence membrane for the detection of hydrogen peroxide. The QD–AF membrane had a linear detection range from 0.1 to 1.0 mM H_2_O_2_ with an LOD of 0.016 mM, and from 1.0 to 10 mM H_2_O_2_ with an LOD of 0.058 mM. The HRP–QD–AF membrane showed higher sensitivity than the QD–AF membrane, although both H_2_O_2_-sensing membranes had good selectivity. The HRP–QD–AF membrane showed possibility to determine concentrations of H_2_O_2_ in the treatment of wastewater containing H_2_O_2_ with highly precise recovery, while the QD–AF membrane could be used as a new analytical method to detect α-ketobutyrate at low and high concentrations.

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
