# Peer review of "Development of Ratiometric Fluorescence Sensors Based on CdSe/ZnS Quantum Dots for the Detection of Hydrogen Peroxide"

_sensors, 2019, doi:10.3390/s19224977_

Round 1
Reviewer 1 Report
This manuscript describes a novel fluorescent sensor HRP-QD-AF membrane for the detection of H2O2, and its application under complex environments. Remarkably, the author demonstrated the reaction of H2O2 with sensor in detail, and eliminated interference from other oxidizing agent. It is a topic of interest to the researchers in the related areas. In addition, in this manuscript, the discussion is logically organized, and each point is cogently developed. Thus, I recommend publication of this work after answering the following problems. Line. 75 It still needs a brief description for synthesis of CdSe/ZnS QDs, although the corresponding references were given. Line. 169 ~ 173 How to calculate quantum yield by relative method or instrument measurement? The corresponding descriptions are lost in the paper. Line. 180, Fig. 1a and 1b The resolution is lower and the scale bar in photo is obscure. Line. 257 ~ 260 and Line. 362 ~ 365 Why the fluorescent intensity of the sensors increase with increasing concentrations of other oxidizing agent except H2O2? The author considered that the fluorescent intensity of AF decreased little in the presence of oxidizing agents. However, the above conclusion lacks of adequate data, such as fluorescent spectrum, or supporting literatures.
Author Response
Reviewer 1
This manuscript describes a novel fluorescent sensor HRP-QD-AF membrane for the detection of H2O2, and its application under complex environments. Remarkably, the author demonstrated the reaction of H2O2 with sensor in detail, and eliminated interference from other oxidizing agent. It is a topic of interest to the researchers in the related areas. In addition, in this manuscript, the discussion is logically organized, and each point is cogently developed. Thus, I recommend publication of this work after answering the following problems.
Line. 75 It still needs a brief description for synthesis of CdSe/ZnS QDs, although the corresponding references were given.
Replyà Thank you for your suggestion, we have added more information for the synthesis of CdSe/ZnS QDs as follows
Page 6,7: “Cadmium acetate dehydrate (0.6mM, 147 mg) and stearic acid (2.13mM, 607 mg) were loaded into a 50 ml three-neck flask and heated to 150 0C under vacuum conditions until a colorless liquid was obtained. Hexadecylamine (1.94g) and TOPO (2.2g) were added to the flask after cooling to room temperature. The mixture was then degassed with a pump and heated to 120-150 0C under vacuum. The reaction vessel was then filled with nitrogen gas and heated to 310-320 0C and, at this point, a solution of selenium (211 mg) in TOP (2.5 ml) was rapidly injected into the vigorously stirred reaction mixture. The solution was heated for 25 seconds before removing the flask from the heating mantle and then allowing it to cool to room temperature. The resulting CdSe nanoparticles were purified by dissolving the reaction mixture in chloroform, followed by precipitation with an equal volume of methanol. In the next step, these purified CdSe particles were used to synthesize the CdSe/ZnS core-shell QDs. A mixture of hexadecylamine (2g) and TOPO (2.5g) were loaded into a 50ml three-neck flask and then degassed and heated to 180 0C. At 180 0C, the purified CdSe particles dispersed in 2.0 ml chloroform were added to this solution. After the chloroform was completely pumped out, the flask was filled with nitrogen gas. The temperature of the reaction was then increased to 180-185 0C. A mixture of zinc acetate (54mg) and (TMS)2S (0.05 ml) dissolved in 1.0ml TOP was then injected dropwise for 5-10 minutes. After the injection, the mixture was stirred for 1h at 180-185 0C.”
Line. 169 ~ 173 How to calculate quantum yield by relative method or instrument measurement? The corresponding descriptions are lost in the paper.
Replyà We have added the method for calculation of fluorescence quantum yield of QDs as follows
Page 7: “Fluorescence quantum yield of the synthesized CdSe/ZnS QDs was calculated by the following equation [30], where rhodamine 6G was used as standard reference.
where F is the area under the emission spectrum, f is absorption factor at a given excitation wavelength of sample or standard, n is refractive index of solvent, and Φ is fluorescence quantum yield. The subscripts, ‘x’ and ‘std’ represent sample and standard.”
Line. 180, Fig. 1a and 1b The resolution is lower and the scale bar in photo is obscure.
Replyà We have added original photos in Fig. 1a and 1b as follows
Page 13:
Line. 257 ~ 260 and Line. 362 ~ 365 Why the fluorescent intensity of the sensors increase with increasing concentrations of other oxidizing agent except H2O2?
Replyà The fluorescence intensity of the sensors increased with increasing concentration of other oxidants except H2O2 because of the pH of solutions. In this work, H2O2 is used as an oxidant, therefore its solution is an acidic medium. Solutions of NaClO, Chloramine T and KMnO4 are fairly alkaline or alkaline solutions, whereas the fluorescence intensity of CdSe/ZnS QDs is usually increased at high pH solutions.
The author considered that the fluorescent intensity of AF decreased little in the presence of oxidizing agents. However, the above conclusion lacks of adequate data, such as fluorescent spectrum, or supporting literatures.
Replyà Fig. 2a indicated that the fluorescence intensity of AF changed little in the presence of H2O2. Moreover, based on data collected from the response of the sensors with three other oxidants we have got this conclusion in the manuscript (page 19). However, as your suggestion, addition of more spectra of these three interfering compounds (e.g. addition of fluorescence spectra into each Fig.3 or Fig.8) to make clearly this conclusion may not be necessary and confusing eyes. Therefore, we have removed this sentence out of the manuscript to avoid questions.

Reviewer 2 Report
It is better to have a scheme to show how detection is processed. Otherwise, it is difficult to understand. The following description is strange. How is it possible that their fluorescence emission spectra did not change significantly but 50% decrease of QY.
“After ligand exchange with carboxyl groups (MPA) on the surface of the synthesized CdSe/ZnS QDs, their fluorescence emission spectra did not change significantly. However, the QY of MPA-coated CdSe/ZnS QDs was decreased to about 50% of its initial value”.
The fluorescence quenching of QDs in the presence of H2O2 is not well-studied. Author should design experiments to show the reaction mechanism. HRP reacts with H2O2. HRP-QD-AF membrane consumes more H2O2. So less H2O2 can react with MPA-coated CdSe/ZnS QDs.
Author Response
Reviewer 2
It is better to have a scheme to show how detection is processed. Otherwise, it is difficult to understand. The following description is strange. How is it possible that their fluorescence emission spectra did not change significantly but 50% decrease of QY.
“After ligand exchange with carboxyl groups (MPA) on the surface of the synthesized CdSe/ZnS QDs, their fluorescence emission spectra did not change significantly. However, the QY of MPA-coated CdSe/ZnS QDs was decreased to about 50% of its initial value”
Replyà We have added a scheme illustrate the detection process in page 5 as follows
Scheme 1. Fabrication of the QD-AF membrane and the HRP-QD-AF membrane and their colorimetric and ratiometric fluorescent response to different H2O2 concentrations
The defects in the ZnS shell after ligand exchange on the surface of CdSe/ZnS core/shell QDs (e.g. defects produce new non-radiative recombination sites) is reason for the decrease of fluorescence quantum yield. We have changed “their fluorescence emission spectra” by “their fluorescence emission wavelength” to avoid misunderstanding.
The fluorescence quenching of QDs in the presence of H2O2 is not well-studied. Author should design experiments to show the reaction mechanism. HRP reacts with H2O2. HRP-QD-AF membrane consumes more H2O2. So less H2O2 can react with MPA-coated CdSe/ZnS QDs
Replyà Fluorescence quenching of QDs in the sensing membranes in the presence H2O2 already showed in almost Figures of this manuscript. This indicates the high response of QDs to H2O2 through fluorescence quenching effect. In addition, this work focuses on the fabrication of the sensing membranes, therefore, we don’t show experiments of QDs probes with H2O2.
The reaction mechanism of QDs with and without HRP to H2O2 is already mentioned in page 16, 23,29.

Reviewer 3 Report
This paper by Hong Dinh Duong and Jong Il Rhee constitutes an interesting original-paper about a development of a highly selective ratiometric fluorescence sensors for hydrogen peroxide and their potential application. The article seems to be very useful and contains a high volume of various and sufficiently detailed information about the sensor, such as synthesis, selectivity, sensitivity, stability, and detection limits. Furthermore, the effects of pH and temperature on hydrogen peroxide determination were investigated. The amount of work is extensive. This is a well-organized manuscript that after minor revision can be accepted for publication. My comments and advices are shown below. They have to be taken into consideration.
The method for detection of hydrogen peroxide was not validated. The results for quantitative determination of H2O2 (not only LOD values) should be compared with at least one other commonly used technique. In other words, the analytical usefulness of the proposed method is not demonstrated compared to other analytical methods. There are no LOQ values, only LOD. Why? LOD values are very low in comparison with the linear ranges (in one case almost 15 times lower than the lowest value on the linear range). It puzzles me. How did authors obtain the LOD values? Line 79. “spectrophotometer” instead of “Spectrophotometer”. Line 143. “ketobutyrate” instead of “keotbutyrate”. It is essential to expand in the introduction section a paragraph on recently developed fluorescent probes used for detection and determination of hydrogen peroxide. For some who are not experts in the field, it would be an advantage to be able to see this in a more comprehensive view. Authors should mention here with appropriate references some more (recent, for example from the last 3 years) papers, among others:
-Chen, Yuzhi, et al. Analytical chemistry 89.10 (2017): 5278-5284;
-Liu, X. et al, Sensors and Actuators B: Chemical, 255, 1160-1165;
- Żamojć K et al. Free radical research 51.1 (2017): 38-46.
- Purdey, et al. Sensors and Actuators B: Chemical 262 (2018): 750-757;
-Zhuang, Zhiyuan, et al. Journal of Photochemistry and Photobiology A: Chemistry 344 (2017): 8-14.
Author Response
Reviewer 3
This paper by Hong Dinh Duong and Jong Il Rhee constitutes an interesting original-paper about a development of a highly selective ratiometric fluorescence sensors for hydrogen peroxide and their potential application. The article seems to be very useful and contains a high volume of various and sufficiently detailed information about the sensor, such as synthesis, selectivity, sensitivity, stability, and detection limits. Furthermore, the effects of pH and temperature on hydrogen peroxide determination were investigated. The amount of work is extensive. This is a well-organized manuscript that after minor revision can be accepted for publication. My comments and advices are shown below. They have to be taken into consideration.
The method for detection of hydrogen peroxide was not validated. The results for quantitative determination of H2O2 (not only LOD values) should be compared with at least one other commonly used technique. In other words, the analytical usefulness of the proposed method is not demonstrated compared to other analytical methods.
Replyà Could you recommend a commonly used technique for H2O2 detection?
Data of Table 1 in this manuscript is somewhat compared with results of other works in the same field.
There are no LOQ values, only LOD. Why? LOD values are very low in comparison with the linear ranges (in one case almost 15 times lower than the lowest value on the linear range). It puzzles me.
Replyà As you know the meaning of LOD is itself name (limit of detection), therefore it is apparent if values of LOD are lower than the first value of the linear detection range.
How did authors obtain the LOD values?
Replyà LOD calculation is based on the ratio of standard deviation (SD) of blank samples and slope of calibration curve (S) as follows:
LOD = 3.3*SD/S
Line 79. “spectrophotometer” instead of “Spectrophotometer”.
Replyà We have corrected this letters according to your comment
Line 143. “ketobutyrate” instead of “keotbutyrate”.
Replyà We have corrected this letters according to your comment
It is essential to expand in the introduction section a paragraph on recently developed fluorescent probes used for detection and determination of hydrogen peroxide. For some who are not experts in the field, it would be an advantage to be able to see this in a more comprehensive view. Authors should mention here with appropriate references some more (recent, for example from the last 3 years) papers, among others:
- Chen, Yuzhi, et al. Analytical chemistry 89.10 (2017): 5278-5284;
- Liu, X. et al, Sensors and Actuators B: Chemical, 255, 1160-1165;
- Żamojć K et al. Free radical research 51.1 (2017): 38-46;
- Purdey, et al. Sensors and Actuators B: Chemical 262 (2018): 750-757;
- Zhuang, Zhiyuan, et al. Journal of Photochemistry and Photobiology A: Chemistry 344 (2017): 8-14;
Replyà We have added these references in the manuscript as Ref. 11, 12, 13, 14, 15 in Section Introduction.
Page 3: “Many fluorescence probes have showed strong affinity to react with H2O2 and they have applied for H2O2 detection in living cells [11-15].”

Round 2
Reviewer 2 Report
none